# Evaluating a Mobility Service Application for Business Travel: Lessons Learnt from a Demonstration Project

**Alfred Andersson [1,2,]\***[ID]**, Lena Winslott Hiselius [1]**[ID]**, Jessica Berg [3]**[ID]**, Sonja Forward [3] and Peter Arnfalk [4]**

1   Department of Technology and Society, Lund University, Box 118, 221 00 Lund, Sweden; lena.hiselius@tft.lth.se
2   K2—The Swedish Knowledge Centre for Public Transport, Bruksgatan 8, 222 36 Lund, Sweden
3   Swedish National Road and Transport Research Institute VTI, 581 95 Linköping, Sweden; jessica.berg@vti.se (J.B.); sonja.forward@vti.se (S.F.)
4   The International Institute for Industrial Environmental Economics, Lund University, 221 00 Lund, Sweden; peter.arnfalk@iiiee.lu.se
*   Correspondence: alfred.andersson@tft.lth.se

**Abstract:** Business travel contributes to significant greenhouse gas emissions, and there is a need for measures that reduce the demand for trips made with energy-intensive means of transport. In this study, a mobility service application (MSA) introduced in 13 Swedish organisations was tested and evaluated to facilitate booking and handling of business trips, in particular public transport. A before and after study consisting of surveys and interviews with employees at the organisations were conducted. The results show that the MSA was mostly used for regional and local public transport trips, and the users stated that the MSA made it easier to travel by public transport, although this particular result should be seen as tentative due to the small sample size. Three factors that influence the success of a new MSA as a means to increase sustainable business trips were identified: management control and proactiveness; perceived improvement of intervention; functions and technical sufficiency. The results also highlight the need to establish organisational conditions that facilitate sustainable business travel, such as coherent travel policy, accessibility to sustainable modes of transport, and a culture that encourages environmentally friendly behaviour. The study suggests improvements that can be made to similar interventions and strategies that can be introduced to promote sustainable business travel.

**Keywords:** business travel; mobility service application; ITS; public transport; mobility management; before-after study

## 1. Introduction

Fossil fuel use is a primary contributor to human-induced carbon dioxide emissions, which aggravate global climate change [1]. Transport is responsible for almost 25% of global energy-related greenhouse gas emissions [2]. In Sweden, this share is even higher (33%) primarily because electricity generation and heating in Sweden is less dependent on fossil fuels [3] Both in Sweden and globally, transport is increasing its share of emissions [2,4]. At the beginning of 2018, the Swedish climate policy framework came into force, stating that by 2045, Sweden will have net zero emissions of greenhouse gases into the atmosphere and should thereafter achieve negative emissions. A separate target was set for the transport sector, declaring that emissions from domestic transport, excluding domestic aviation, will be reduced by at least 70% by 2030 compared with 2010 [5]. Hence, promoting more sustainable

and energy-efficient travel behaviour is of substantial interest, and there is mostly a consensus among transport researchers on the need for levels of transport to be reduced in order for the sector to contribute to more sustainable development [6,7]. In order to reach these ambitious targets a number of measures need to be introduced including changing transport behaviour [8,9].

Organisations generate a great deal of business travel globally. In Sweden, according to the latest national travel survey (RVU Sweden, 2011–2014), business trips account for 10% of the total number of passenger kilometres travelled per person and day [10]. Business trips by air have mainly been in focus within media as part of the "flight shame movement" while business trips by car have gained less interest even though they constitute a larger share of total passenger car kilometres per person and day (9% according to RVU Sweden). Further, local and regional public transport authorities have rarely prioritised business travel in their plans though a majority of trips carried out are over short distances less than 10 km and 40% are shorter than 5 km [10].

Thus, there is a great and untapped potential to contribute to the policy objectives of long-term sustainability in the transport sector by reducing the number of business trips made by cars. Various transport policy measures are available to reduce people's car use and to increase the use of more sustainable transport modes [11]. Some of these are referred to as 'soft' measures, that focus on voluntary changes such as, campaigns, travel plans for organisations or free public transport tryouts. Such measures aim to motivate individuals to voluntarily change their modes of transport to more sustainable ones [12] and have been implemented in several countries including Sweden [13]. Recently, soft measures have been studied in combination with information and communication technologies (ICT) to further promote a shift away from private car use [14,15]. While many studies focus on commuting trips [16], previous research has not sufficiently explored the role of business travel in the transition to a more sustainable transport system. Research on travel behaviour in workplace intervention contexts is needed to advance the understanding of how sustainable business trips can be facilitated in practice [17].

Further, the ongoing digitalisation has opened the possibility to combine mobility services with ICT to create a package of mobility solutions. Currently, there is a trend of new forms of shared mobility services (referred to as mobility as a service or MaaS) being developed to facilitate a multimodal and sustainable travel behaviour by reducing the need to own a private car [18]. Several evaluations have been made on these services concerning individual travel behaviour to explore its implementation issues and potential sufficiency in replacing car journeys [19,20]. The use of ICT also offers the possibility to avoid the need to travel through the use of digital applications providing virtual access to work, meetings, healthcare, education, etc., in what is referred to as accessibility as a service or AaaS [21]. Virtual meetings are ICT-enabled accessibility services that substantially can, given the right conditions, reduce the need for business travel [22,23]. However, recent studies have questioned whether mobility services such as MaaS will be able to decrease private car use [24], and suggested that expectations might be inflated [25], demonstrating the need for more research investigating the sustainability of these new services. Moreover, few studies have investigated the implementation of mobility services in organisations to promote more sustainable business trips.

The current study aims to fill in this research gap by demonstrating a new mobility service application (MSA) for business trips in Sweden and to evaluate it within the context of organisational travel management and practices. Survey data and interviews were used to analyse participant's travel behaviour change, perceptions of business trips, how they think their organisation manages such trips, and what they thought of the MSA as a support to facilitate more sustainable business trips.

This paper is organised as follows: Section 2 includes background and summarises previous research on business travel and mobility services. Section 3 introduces the analytical and theoretical framework. Section 4 describes the methodology, research design and design of the study. Section 5 presents the results of the survey and interviews separately, and Section 6 discusses these results. The study's conclusions are presented in Section 7.

## 2. Background

### 2.1. Business Travel Behaviour

Business travel can be defined as people travelling for work-related purposes. Davidson and Cope [26], divide business travel into individual business travel, which comprises the regular trips necessary to carry out employment tasks; and business tourism which includes a variety of business meetings and events and is sometimes associated with MICE (meetings, incentives, conferences and exhibitions) industry. Business travel does not only relate to the individual traveller's behaviour and conditions, but also to policies and the organisational culture around business travel. Hence, theories and models for individual travel behaviour, such as the theory of planned behaviour and the transtheoretical model, cannot easily be applied.

Business travel is commonly regulated in a corporate travel policy with associated guidelines regarding travel and its administration. Business travel is often managed through business travel agencies providing similar services to traditional travel agents, such as making reservations, issuing tickets and providing advisory services. However, even if a travel policy is present, research has shown that employees enjoy relative large freedom, particularly management, to decide on whether to take a trip and by what travel mode [27].

According to Gustafson [28], developing and implementing a travel policy is a cornerstone in an organisation's effort to control its travel activity. The main objective of the travel policy is to establish common rules and routines. It contains regulations on how to travel, what means of transport and what suppliers to use, what degree of comfort is allowed (e.g., economy or business class), what kind of ticket to use, and so forth. The travel policy also specifies what administrative routines travellers should follow, such as pre-trip approval, booking procedures, payment routines and expense report management.

Travel policies often deal with guidelines regarding virtual meetings such as audio-, web-, and videoconferencing as well. These alternatives have shown to be a useful measure for reducing the environmental load. An active investment in increasing the proportion of virtual meetings in 19 government agencies in Sweden (REMM—virtual meetings in authorities) resulted in an average reduction in $CO_2$ emission from business travellers per employee by 25% over a seven-year period, which can be compared to other Swedish authorities where corresponding emissions decreased by 6% during the same period [4]. Easily accessible information about the virtual alternatives and a smooth booking process, preferably closely linked to the travel alternatives, is an important success factor [29,30].

There are also some evaluations carried out for business sales activities indicating a potential to increase the number of trips by public transport. The local public transport agency in Stockholm (SL) conducted a follow-up study of a sales activity towards companies where companies were provided with special company tickets to be used in service. The study indicated that of those receiving the company ticket, the share of trips by public transport increased by 27% and the number of trips by car decreased by 20% [31]. In a study by Forward [13], the effect of a free travel pass resulted in a more positive attitude towards bus usage, with a large number having either changed or having started to change their behaviour. When the same people were contacted three months later, 50% still used public transport.

### 2.2. Mobility Services

New, smart mobility solutions for business travel can create better conditions for organisations to contribute to sustainable transport policy objectives, by providing incentives to travel by public transport, walking and cycling, or having virtual meetings. Many actors in Sweden and internationally are now highlighting "mobility as a service" as a priority area for developing a more sustainable transport system. Some experiments with integrated mobility services, mainly aimed at private travellers, have been carried out. Karlsson, Sochor and Strömberg [32] who studied the effects of a

MaaS field trial in Gothenburg, found that users were generally satisfied with the service and that 48% reported less private car use during the experiment. However, even though these have shown positive results, there are a number of problems that need to be solved if this type of service is to be established. Offering an MSA to facilitate more public transport use does not always succeed in convincing people to do so. A number of studies have shown that it is also important to increase the quality of the service [33].

Barriers and Facilitators

In order to achieve higher quality and satisfaction, the focus should be on how to satisfy the traveller's needs. Previous studies have shown that some of the most important requirements are: ability to buy tickets and be able to integrate different ticket systems, real-time information, and information on the entire route including transfers and personal information [34–37].

An important advantage of an electronic ticket is that it reduces the uncertainty about the waiting time. A further advantage is that it can give the user information about their travel. Dekkers and Rietveld [34], found that 58% liked the fact that their ticket gave them insight into how they travelled and what it costs. According to Link et al. [38], an important obstacle to using public transport is that all the different bus companies had their own, special cards and ticket systems. A single, standard card for the whole of Sweden would simplify travel. This is also supported by Turner and Wilson [39], who argued that an integrated ticket systems could offer greater flexibility and simplicity for passengers.

Provision of real-time information is becoming an increasingly fundamental part of the services offered by public transport companies and is considered to be the most important characteristic [35,40]. Real-time information can help travellers feel more in control, but it can also increase their sense of security [41]. Travellers need detailed information both before and during the trip, and especially when changing. When it comes to disruptions, travellers need clear guidance on how to continue their journey [42].

A trip with many exchanges often involves a high degree of uncertainty [43] and can be an important barrier to travelling with public transport [36,42]. Therefore, the traveller needs information both before and during the trip [33,43]. This information is not only about real-time information but also how to get to the station/stop, what the station/stop looks like, what service is offered and where and how the exchange itself can be done. Preferably, the information should be adapted to individual preferences.

Customised service can provide users with additional support for choosing multimodal transports [37]. Personal travel information is also expected to lead to a reduction in car journeys [44]. Studies have also shown that customised information is something the traveller wants [38]. A problem that may arise in connection with this is an intrusion on personal integrity, but studies have found that this is not a problem as the benefits seem to outweigh the disadvantages [38].

The presentation of information and usability are other important aspects. The customer is demanding a system that is reliable, understandable and easy to read [45]. This means that public transport operators should try to design systems to be as user-friendly as possible [46,47] and do not require a large amount of training and specialisation [46].

## 3. Theoretical and Analytical Approach

To demonstrate and evaluate the implementation of an MSA for business trips, the present study analyses the individual traveller's behaviour and conditions as well as the organisational culture and management practices related to business travel.

For this more comprehensive approach, we use 'the unified theory of acceptance and use of technology' (UTAUT), defined by Venkatesh et al. [46]. The UTAUT aims to explain user intentions to use an information system and subsequent usage behaviour. The UTAUT states that perceived usefulness (performance expectancy), perceived ease of use (effort expectancy) and norms (social

influence) affect technology adoption intention via behavioural intention, which in turn leads to behaviour; whereas facilitating conditions directly antecede behaviour [48]. Table 1 presents the definitions of these determinants.

**Table 1.** Factors that influence technology adoption according to the unified theory of acceptance and use of technology (UTAUT) [49].

| | |
|---|---|
| Performance expectancy | Defined as the degree to which an individual believes that using the system will help him or her to attain gains in job performance. Five constructs that pertain to performance expectancy are perceived usefulness, extrinsic motivation, job-fit, relative advantage, and outcome expectations. |
| Effort expectancy | Defined as the degree of ease associated with the use of the system. Three constructs capture the concept of effort expectancy: perceived ease of use, complexity, and ease of use. |
| Social influence | Defined as the degree to which an individual perceives that important others believe he or she should use the new system. It refers to the way in which individuals change their behaviour to meet the demands of a social environment. |
| Facilitating conditions | Defined as the degree to which an individual believes that an organisational and technical infrastructure exists to support the use of the system. |

The holistic approach in UTAUT fits well to analyse the implementation of an MSA in businesses because it considers both individual and organisational factors that are subject for investigation in this study. The application of the theory in this study is illustrated in Figure 1.

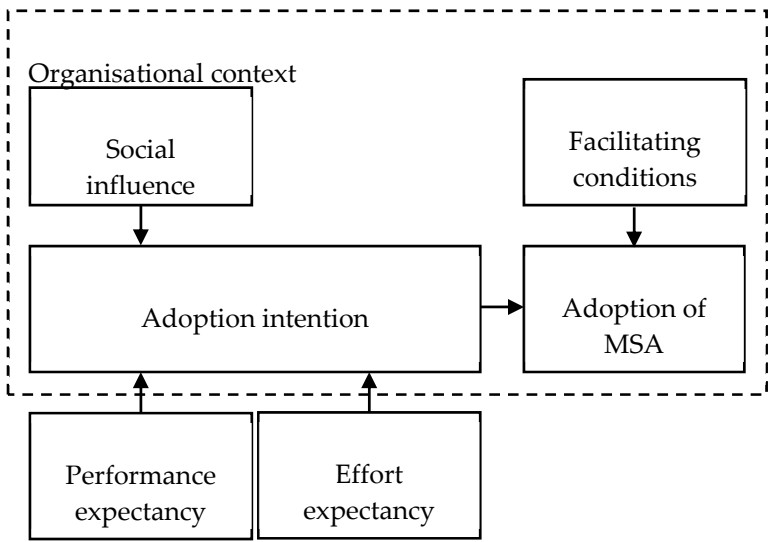

**Figure 1.** The analytical framework used to evaluate the implementation of the mobility service application (MSA).

## 4. Methods and Research Design

Through surveys and interview studies, quantitative data were combined with qualitative data. This mixed-method approach enables an analysis of the context, implementation, design and function of the evaluated MSA.

### 4.1. The MSA

Samtrafiken is owned in equal shares by all regional public transport authorities and most of the commercial public transport operators in Sweden. Samtrafiken connects all public and private transport operators, coordinate public transport data and develops and manage ticketing and payment standards. Over three years (2017–2019), Samtrafiken provided a service consisting of a mobile web application where, among other things, employees of recruited companies, authorities and organisations could manage their business trips by public transport and car. The project was based on the idea that employees should only need one tool regardless of the mode of transport. The MSA provided timetable information as well as the purchase of tickets for local and regional buses, commuter, regional and national train services, registration of car trips, and reporting of travel expenses. Each region had its own outline of the MSA. Through the MSA, the trips by public transport were paid through a monthly invoice including a financial statement from Samtrafiken to the companies, authorities and organisations that were recruited.

Samtrafiken coordinated the project with a reference group consisting of representatives from 6 regional public transport authorities (Hallandstrafiken, Region Kronoberg, Samtrafiken, Stockholm public transport authority, Upplandstrafik and Östgötatrafiken). Through the reference group, users of the MSA were recruited continuously during the first two years of the project. In other words, the researchers who authored this paper did not control the selection of participants. However, throughout the project, the researchers followed the implementation process and evaluated the MSA and its use. In total 13 organisations were recruited and appointed contact persons were provided with information material about the MSA to be distributed in each organisation. Participating actors also provided Samtrafiken with contact information (email addresses) to involved employees, facilitating the evaluation of the MSA.

### 4.2. Recruitment and Procedure

The total number of persons in the initial target group was 525, employed at four companies, eight public authorities and one NGO. Via the Netigate survey tool, a before and after study was carried out in that an email was sent to each participant with a unique link to the web survey. By this approach, targeted reminders can be sent to those who have not answered the survey (wholly or partially). The e-mails for the before study were sent out in two rounds, in March 2017 and during February–June 2018. The web survey was distributed as new participants came to the project's knowledge, hence the extended sending period in 2018. Up to three reminders were sent. Trough participating organisations, information on the MSA was spread via workplace meetings, leaflets and emails. In the after study, an email with a web link was solely sent to those who in the before study stated to make business trips. The mailing was made in mid-April 2019. Three reminders were sent. These individuals were also contacted via email with a request for an interview. Statistics on the number of persons in the target group and the number of responses received in each study are presented in Table 2.

**Table 2.** Number of persons in the target group and responses in the studies.

|  | Study Population | Responses |  |
| --- | --- | --- | --- |
| Survey (Before study) | 525 | 250 (48%) | where of 193 (77%) carry out business trips |
| Survey (After study) | 193 | 77 (40%) | where of 35 (18%) used the MSA |
| Interviews (After study) | 193 | 40 (21%) | where of 20 (50%) used the MSA |

*4.3. Design of Studies*

4.3.1. Survey

The questions in the before and after study were based on two previous studies on travel behaviour [50,51] with some revisions following discussions within the project group. The before-after study consisted of a short travel survey concerning business trips (distance, transport mode, number and frequency of business trips) but also questions regarding attitudes towards various transport modes and perceived possibility to use these modes for business trips. The after study also consisted of questions related to the use of the MSA presented to those stated to have tried the MSA. Of 77 respondents in the after study, 35 respondents used the MSA. Responses from both groups (users and non-users) were analysed in order to analyse the effect of the MSA and for the users, to also gain knowledge on various aspects of the MSA.

In the before-after study, respondents were asked to answer some questions about business trips related to their perceived possibility to use public transport, how easy they think that is for them, and to what extent their near colleagues use public transport. The first question was stated as 'How possible is it for you to travel by bus/train for business trips' (performance expectancy)? The second was a statement saying, 'You think that using the bus/train is difficult' (effort expectancy). The third question was 'Your closest colleagues, who also make business trips, how often do you think they travel by bus/train' (social influence)? Each question was asked once for bus trips and once for train trips, and the scale was from 1 to 7, where 1 was impossible/totally agree/very seldom and 7 was very possible/totally disagree/very often. The mean score was then computed from the bus/train questions to get an average public transport score for each determinant. Histograms were produced to control for the assumption of normality, which indicated roughly normal distribution for effort expectancy and social influence, but the third determinant, performance expectancy, seemed to deviate from this assumption. The Kolmogorov-Smirnov and Shapiro-Wilk tests suggested that the data are normally distributed for effort expectancy ($p = 0.200$ and $0.428$) and social influence ($p = 0.200$ and $0.57$). Conversely, for performance expectancy, both tests were statistically significant ($p = 0.006$ and $0.001$, respectively), indicating a normality violation for that variable.

Attempts were made to normalise the performance expectancy variable using three different transformations (square root, reflected inverse, and log base 10) but neither succeeded to improve the shape of the distribution for the variable. Therefore, it was decided to omit that variable from the analysis and proceed with tests on effort expectancy and social influence. Significance tests were conducted between the before-after study for MSA users and non-users respectively too see whether the MSA influenced these two determinants.

In the after study, the respondents were also asked how often the MSA has been used for different types of business trips using scale 1 = never and 7 = always. In the after study, the users also answered questions about how good the application had been regarding various functions. The functions, graded on a scale from 1 = very bad, 7 = very good, were technology, login, real-time information, invoice handling and information on where to find stops. Finally, questions about virtual business meetings were asked, how they use such meetings today and if it would be useful for them to have that kind of service in the MSA.

4.3.2. Interviews

Ten individual interviews and nine focus groups were conducted (a total of 19 interviews) with a total of 40 people, see Table 3. The definition of a focus group here is when an interview takes place with two or more people. In some organisations, several interviews were conducted. There were 12 organisations represented. For practical reasons, seven interviews were conducted over the phone. On average, an interview took between 1 and $1\frac{1}{2}$ h. The interviews were conducted during February and March 2019. In most cases, the interviews were conducted at the informant's workplace. The recruitment to the interview study was carried out before the after study; thus, it was not possible to

identify the MSA users preceding this. Thus, in the interviews, there was a mix of respondents that used the MSA, tried to use the MSA but did not succeed for any reasons, and non-users of the MSA.

**Table 3.** Statistics on interviews carried out.

| Organisational Level | Number of Organisations | Number of Interviews/Group Interviews | Total Number of Persons Interviewed |
|---|---|---|---|
| Municipal authorities | 3 | 9 | 18 |
| Regional authorities | 2 | 2 | 3 |
| Authorities and organisations on the national level | 2 | 3 | 5 |
| Companies owned by a regional authority | 1 | 1 | 1 |
| Companies owned by a municipal authority | 1 | 1 | 2 |
| Privately-owned companies | 2 | 5 | 8 |
| NGO | 1 | 1 | 3 |

Interviews were conducted to gain a deeper understanding of how business trips are planned and implemented, how temporal and organisational constraints affect opportunities to manage business trips, and the extent to which an MSA meets these needs. Interviews also provide an understanding of how this type of service works for the organisation (both administrative and organisational aspects).

Semi-structured interviews were used to ensure that a number of important issues were covered in all interviews, but also to provide opportunities for respondents to bring up issues the interviewer had not considered [52]. Interviews thus took the form of conversations in which the interviewer asked open-ended questions and follow-up questions within relatively broad pre-defined themes. The following themes were addressed during the interviews:

- Distance to work and means of transport;
- Frequency of business trips, distance and means of transport;
- Travel Policies;
- Virtual meetings;
- Meeting and travel culture;
- Questions about the MSA.

The analysis took a content analysis approach [53]. The interviews were recorded with the informant's consent and then transcribed. After that, the transcripts were coded and analysed thematically. Initial coding mainly used the pre-defined themes from the interviews; subsequent coding and analysis developed themes and sub-themes in an interplay between the empirical data and existing research and theory [54,55]. The most important analytical themes that emerged from this process are presented in the following sections and discussed in relation to the UTAUT.

In the presentation of the results, quotes from the interviews are used to illustrate the respondent's reasoning. These are reproduced verbatim but may have been adjusted for reading comprehension. If the text needs clarification, this has been written in brackets.

## 5. Results

### 5.1. Surveys

#### 5.1.1. Differences between MSA Users and Non-Users

Firstly, we compare various statistics on MSA users and non-users in order to see whether there are any significant differences in the populations that may influence the evaluation of the intervention. According to the before-study, the travel behaviour of the users of the MSA (35 persons) and the

non-users (158 persons) is similar except that the users of the MSA had significantly higher shares of business trips made by train and commuter train, see Table 4. The overall share of public transport for business trips is high (46–44%) compared to statistics from the national travel survey on the number of trips where only 10% of the business trips are made by public transport. The mean age for the MSA users was 48.4 and 44.9 for the non-users. Women were overrepresented in both groups, 71% for MSA users and 61% for the non-users. Both groups had high access to a bicycle pool (83% for MSA users and 69% for non-users) and fairly high access to a car pool (60% for MSA users and 50% for non-users) while only a small proportion in both the groups had their own company car (14% for MSA users and 10% for non-users).

**Table 4.** Independent samples *t*-test to explore potential differences between the groups regarding business travel modal share (number of trips), as stated in the before-study.

| | Car | Car (Passenger) | Bus | Train | Commuter Train | Bicycle | Walking | Other |
|---|---|---|---|---|---|---|---|---|
| MSA users % | 22 | 11 | 12 | 24 ** | 10 * | 11 | 6 | 4 |
| Non-users % | 24 | 14 | 12 | 22 | 8 | 9 | 8 | 3 |

** $p < 0.01$, * $p < 0.05$.

The use of virtual meetings is similar between MSA users and non-users (approximately 75% in both groups state that they are using virtual meetings for business). The majority of both groups have virtual business meetings 1–3 times/week. Further, there were no statistically significant differences between the groups when asked questions regarding their experiences with, and attitudes towards virtual business meetings (Table 5). The result indicates that the technology for virtual meetings is neither good nor bad and that the respondents are slightly positive towards the idea of having the possibility to book virtual meetings in an application. The scores are somewhat lower when respondents were asked to rate the suitability of their business trips to be made virtual, their employer's encouragement to use virtual meetings, and the respondent's perceived ability to arrange such meetings.

**Table 5.** Independent samples *t*-test to explore potential differences between the groups regarding attitudes towards, and experiences with, virtual meetings, as stated in the before-study.

| | We Have Good Technology for Virtual Meetings | We Have Good Technology, But Virtual Meetings Are Not Encouraged by My Employer | My Business Trips Are Suitable for Virtual Meetings | I Feel Confident in How to Arrange Virtual Meetings | It Would Be Useful to Be Able to Book Virtual Meetings in an Application |
|---|---|---|---|---|---|
| * MSA users | 4.22 | 4.22 | 3.11 | 2.56 | 4.19 |
| * Non-users | 4.71 | 3.71 | 3.65 | 3.43 | 4.44 |

* 1 = Do not agree at all, 7 = Fully agree.

### 5.1.2. Changes in Attitudes and Perceptions Towards Public Transport

As shown in Table 6, the only significant difference was for the MSA users in relation to effort expectancy, suggesting that they perceived it easier to use public transport after the use of the MSA. However, due to the small sample size, caution should be taken to generalise these results.

**Table 6.** Paired samples *t*-test to explore potential differences between the before and after study for MSA users and non-users.

| | MSA Users | | Non-Users | |
|---|---|---|---|---|
| Determinant | Before | After | Before | After |
| [a] Effort expectancy | 4.57 | 4.94 * | 4.66 | 4.76 |
| [b] Social influence | 4.14 | 4.54 | 3.98 | 4.49 |

* $p < 0.05$, [a] 1 = totally agree, 7 = totally disagree, [b] 1 = very seldom, 7 = very often.

### 5.1.3. Use of the MSA

The analysis of information from the before-after study reveals that there was no significant change in modal share when using the MSA. This may be due to the relatively high use of the respondents using public transport from the beginning but also due to infrequent use of the MSA. According to results, the application was only sometimes used for trips within the region (average 3.09) and rarely for trips to another region (average 2.32) and within the business location (average 2.39). Regarding the use of the MSA, the after study included questions regarding what type of trips the MSA had been used for (multiple answers were possible). The response options consisted of local bus journeys, regional trains/bus journeys and long train journeys. Almost 70% had used the application for regional train/bus trips. In total, 45% used the MSA for local bus journeys while 25% used it for long-distance train journeys.

### 5.1.4. Perception of the MSA

The result indicates that the MSA was perceived as neither good nor bad. Characteristics on the MSA scored somewhat higher (technology 5.25 and login 5.12) compared to aspects regarding information and administration (real-time information 4.89, invoice handling 4.71 and information regarding stops 4.19). Worth noting is that these are the answers from the group that used the MSA. Those who tried to use the MSA but cancelled due to problems are not included.

### 5.1.5. Summary

The result from the surveys indicates that the MSA was just tested occasionally, mainly by users who already had a fairly high use of public transport. The MSA users and the non-users had similar experiences and attitudes from using virtual meetings, and both groups indicated that it might be useful to incorporate virtual meetings in a future travel application. The results show that the MSA users thought it was less effortful to use public transport after they had experienced the application. A similar difference could not be found in the non-user group. The trips made using the MSA consisted mainly of regional trips by train and bus. The functionality of the MSA was graded as satisfying (neither good nor bad) by the users. In the end, these results only give us a vague picture of the employees' need and use of an MSA for business trips due to the limited use of the MSA. In order to gain a deeper understanding, we now turn to the result based on the interview study.

### 5.2. Interviews

From the interview material, a number of barriers and facilitators that affect the uptake of sustainable business trips in general, and an MSA in particular, were identified. In analysing these findings, three themes were developed: (1) management control and proactiveness; (2) perceived improvement of intervention, and; (3) functions and technical sufficiency. The following sections examine more closely the contribution of these themes in light of the UTAUT theory.

### 5.2.1. Management Control and Proactiveness

Several respondents reveal that their organisations encourage sustainable business trips by promoting the use of bicycles and public transport. The fact that there are available alternatives that are easy to choose is an important reason why sustainable business trips take place. One of the municipalities that participated in the study offers electric bicycles to their employees and has also given the opportunity to try an electric bicycle and a public transport travel card for private trips. In another organisation, campaigns have been implemented to reduce car commuting by raising parking fees and reducing the number of parking spaces. In several organisations, guidelines are saying that employees must use rental cars or pool cars that run on renewable fuels for business trips instead of using their private car. Common to most of these organisations is that they have developed

and implemented a travel policy designed to steer towards a vision or long-term goals, often related to reduced carbon dioxide emissions.

It is common that the travel policies implemented in the studied organisations instruct the employees not to fly domestically, that trips should be made by public transport and that one should question whether one needs to travel at all. However, the respondents state that in many of the studied organisations, the travel policy is unknown to the employees, or has weak support and is considered unclear. One interviewee state that the travel policy is not clear and tough enough to lead to sustainable business travel in practice:

> Sometimes it is hard to argue why you want to take the train to Europe, even if you are prepared for it to take longer.
>
> (male employee at municipal authority).

Several of the respondents mention that time is given priority over choosing the most environmentally friendly way of travelling. For instance, a manager in a municipal organisation thinks that it is the HR department's responsibility to develop a travel policy. On the other hand, another respondent responsible for the travel policy at her organisation explains how important it is for management to follow the guidelines:

> There was a suggestion that everyone would walk, cycle or ride a bus, which the management thought was good, but then only one of them complied with the proposal. This [the management] is a very important group. Unless the management is involved, it will not work. It feels frustrating. It gets lonely.
>
> (female employee in an organisation on the national level).

A travel manager at a company says employees need to understand why it is important to make business travel more sustainable for it to happen. He further believes that the management must lead by example and implement clearer guidelines that are followed up continuously:

> It is not enough to say that now everyone should use electric cars, but you have to understand why, you have to build this common understanding of who we want to be and what kind of society we want to contribute to.
>
> (male employee at a company).

There are also examples on the opposite. In one of the municipalities, there is a green travel plan that works to stimulate sustainable travel. Some employees in the municipality mention that the management team talks a lot about the plan and about using public transport. Another municipality is discussing whether it is worth attending some of the meetings to better utilise their time and resources. If participation is necessary, the opportunities to participate online should be considered according to the travel policy.

How business trips are communicated and managed in workplaces has an impact on the organisational culture. Organisations with coherent travel policy and management attitude seem to have a more streamlined organisational culture with regards to sustainability and business trips. In organisations with less management control and proactiveness, respondents report that attitudes diverge considerably between co-workers on these issues. This could have a significant influence on behaviour. According to the UTAUT theory, social influence affects an individual's behaviour, especially in mandatory settings, where the compliance mechanism is particularly influential [48]. In an interview with two respondents working at a regional authority, several examples are given on how car use is indirectly encouraged by management; managers drive in their cars, the organisation's conference facility is only reached by car for "getting away from work", and free parking is available at the workplace. It is also considered advantageous to travel in your car. Higher mileage allowances than the Swedish tax agency's recommendations promote driving a car. Furthermore, there are advantageous

car company contracts. The car is also considered to be a status mark in some organisations, and it appears that the higher up the hierarchy one gets, the better the company car.

> If you do not have a company car, you have mileage allowances that many see as an extra income to finance the private car.
>
> (male employee at a company B).

> . . . but it also has no consequences when you take the car, no one says anything, no one even says that "we have a travel policy".
>
> (female employee at company A).

Despite the apparent discrepancy in some organisations between their policies for business trips and how trips take place in practice, there are several examples of strategies that seem to have a positive effect from a sustainability point of view. Developing and implementing a travel policy which is anchored with senior staff members and management is one such strategy. Another is to make sustainable business trips more viable through incentives that promote cycling and public transport (green travel plan, pool cars, easy access to e-bikes and public transport tickets), and disincentives for car trips (less parking and higher fees, a prohibition to use the private car for business trips, etc.). Finally, the respondents suggested that it is crucial to create an organisational culture that promotes sustainable business trips to facilitate social norms that reward pro-environmental behaviours.

### 5.2.2. Perceived Improvement of the MSA Intervention

The respondents with experience from the MSA (tried or used) generally had a mixed opinion on whether the MSA made it easier to make more sustainable business trips compared to the current system. The perception of improvement that people relate to the new system is essential for a successful diffusion. According to Venkatesh et al. [46], performance expectancy is the strongest predictor of intention and the higher the perceived relative advantage of the innovation is compared to the idea it supersedes, the more rapid its rate of adoption is likely to be.

Generally, respondents were positive about the idea with an MSA or a similar system as a support for increasing sustainable business trips, regardless of whether they had positive or negative experiences with it. It was particularly desirable to have one system that offers all means of transport, in line with the MaaS-concept.

> It would be great if you could book a rental car in the app, book a taxi and a bicycle. More like a travel app where you can pay for all types of transport. That would be neat.
>
> (female employee at a privately-owned company).

At some workplaces, employees had to borrow a joint travel card for business trips with public transport. In these cases, employees thought it was particularly convenient to have the MSA instead, although one shortcoming was that there were too few ticket-options which sometimes led to more expensive trips than would be the case with the company's travel card. Still, the fact that the MSA bundled travel expenses automatically was perceived as a significant improvement for many respondents compared to the current system where employees must claim reimbursement afterwards for each business trip.

> Yes, it has saved me time. I do not need to report every [business] trip but can do it in lumps.
>
> (female employee at a privately-owned company).

The fact that the application can save time on the individual level is crucial because it can further be perceived to enhance job performance, which has been shown to significantly increase intention [48]. However, one respondent thought that the administrative benefits on the organisational level should

have been shown to the employees as well to increase the general understanding of why the new system was implemented.

Larger organisations usually have a procured travel agency that handles business trips for employees. Some respondents indicated that the travel agency they were already using was working relatively well, and consequently, they saw less value with the MSA because they thought they already had a sufficiently good business travel booking system. Some stated that they did not use the MSA because they thought they had to book business trips through the procured travel agency. The least need for the MSA was for those who usually walked or biked to business meetings. This shows the importance of relating to current systems and the needs of different users.

> . . . [the MSA] can become far too complex if all wishes are to be taken into account. It can then be difficult to use.
>
> (focus group interview within an organisation on the national level).

### 5.2.3. Functions and Technical Sufficiency

Many of the respondents had comments related to the function of the MSA; its technical sufficiency, and the support received/not received to alleviate technical issues. The comments were primarily related to either purely technical weaknesses such as login problems, or functional deficiencies such as lack of ticket options. According to UTAUT theory, such issues are linked to effort expectancy and affects intention to adopt new interventions [48].

As previously mentioned, respondents appreciated that the MSA handled travel expenses. However, the aggregated invoice that was compiled at the end of each month was problematic for the administrative staff and accountants, who found it challenging to match trips with respective projects. Functions that some respondents thought missing were the possibility to book an overnight stay in combination with the trip, book tickets for a group of people, and more ticket options in general. Another issue was that purchased tickets came in the format of a text message and this unusual format made drivers suspicious of whether the ticket was real or not. One interviewee who thought that the MSA was useful in general, complained about the requirement to report administrative information before the ticket purchase:

> When I buy a ticket, I must enter the customer number and project cost centre. It's okay, but I have to do it before I buy the ticket. I am often on the move and often forget to buy the ticket on time so when I am at the station, I must remember the customer number and the cost centre. ... It would be great if I could put it in after I bought the ticket.
>
> (female employee at a privately-owned company).

Several respondents experienced technical weaknesses with the MSA. The interface was not like that of a real smartphone application, but more like a website. It was perceived to be slow, and some respondents had to log in every time which was time-consuming. The login procedure was too complicated and quickly forgotten if business trips were not made very often. When these problems arose, some respondents perceived that they did not get enough technical support. Despite these limitations, respondents were generally positive towards the MSA and maintained that there is a need for a better system than the current solutions.

> . . . I believe in these kinds of trials ... but as the MSA looks right now it needs to be improved before it can be used on a large scale.
>
> (employee at a non-profit organisation).

Although it seems obvious that developers of MSA:s should take functions and technical sufficiency seriously, it seems that the development of increasingly sophisticated MSA:s (both in terms of functions and interface) also leads to higher expectations and demands from users. The respondents often

compared the evaluated MSA with existing travel applications provided, for example, by the national passenger train company SJ, with comments such as: 'The MSA must be easy to use, much like SJ's app'. Thus, a better concept for business trips must also be combined with a user-friendly interface and technical sufficiency.

### 5.2.4. Summary

In summary, the interviews highlighted the importance of facilitating organisational conditions that favour sustainable business travel. Management needs to take responsibility for implementing and anchoring a travel policy, making sustainable transport accessible, and creating an organisational culture that encourages pro-environmental behaviour, and lead by example. The respondents who used the MSA were generally positive to it as a means of managing business travel. The automatic handling of travel expenses was especially appreciated, but there were also shortcomings (technical and functional) that prevented use as well as wishes to include a greater range of transport services. The results also show that an MSA needs to be integrated with existing systems and guidelines to avoid conflicts that might otherwise occur to the user.

## 6. Discussion

This study aimed to evaluate a new MSA for business trips within the context of organisational travel management and practices. The findings suggest that participants were generally positive towards the MSA and that there is potential to improve conventional systems managing business trips. Still, there is a heterogeneity aspect of users and actors to consider when designing interventions promoting sustainable business trips. Different levels of actors influence practices related to business travel, and our findings demonstrated the importance of involving the management in the facilitation of travel policies, travel cultures, and other facilitating conditions such as convenient accessibility to sustainable modes of transport and discouragement of unsustainable ones. New MSAs must be compatible with existing systems within the business travel practice, be well functioning and perceived as an improvement compared to the conventional system, to be utilised. This study also demonstrates some methodological challenges with evaluating new mobility applications. We discuss these issues in more detail below.

### 6.1. Evaluating the MSA

The survey results indicated that effort expectancy got more favourable for MSA users, providing tentative results that the intervention could have increased the ease associated with the use of sustainable business trips. This insight was reinforced by the interviews where several respondents stated that the MSA made it easier to travel by public transport, partly because of the automatic handling of travel expenses. At the same time, many stated that the MSA had several shortcomings that need to be addressed to make it competitive. In addition to the purely technical aspects, the MSA was expected to offer more ticket options and be easier to use. An important reason for not using the MSA was that existing business travel booking systems were either procured and thus employees had to use that service, or that existing booking systems were more flexible and offered more personalised service. Earlier research has stressed the need to adjust interventions to the need of the user and to contextualise content to make it more relevant to the user [56–58]. While it often is more practical to develop one system for all, previous research has stressed the need to adapt systems for users with diverging needs and expectations in order to increase the uptake of interventions for sustainable mobility [14]. Some respondents in this study felt that the current booking system has an advantage in that it offers personalised service, which includes not only the trip itself but also the booking of overnight stays. Should problems arise, it can be felt like security to be able to contact a booking manager who can solve the situation. However, several pointed out that the current system also had its shortcomings and that there was a potential to introduce a booking system similar to the evaluated MSA, especially if such a system made it possible to book several different types of transport services,

and also accommodation. From the responses to the survey to judge, there also seems to be reasons to include the possibility of booking virtual meetings in such an MSA.

There are many factors that play a role in the choice of transport, which is hardly controlled by just one application. However, it seems that the current business travel management system lacks features that make it easy to make sustainable travel choices. Thus, there should be potential for new systems that are designed in line with the user's preferences and needs, while at the same time facilitating sustainable business travel.

*6.2. The Crucial Organisational Context*

From the interviews, it became clear that business travel is produced within an organisational context that differs significantly between organisations. Travel policy, access to travel modes and to virtual meetings, and organisational culture are all contextual factors beyond the individual's direct influence when deciding whether to go travelling or not, or choosing means of transport for a business trips. These factors are determined by the management, who was often referred to by employees as a cause for weak compliance to sustainable business trips. Although the majority of organisations had a travel policy, many respondents claimed that employees either did not follow it or knew it existed and that this was often ignored by management. The important role of managers in promoting sustainable business trips have been stressed in earlier research. Gustafson [26] found that travel managers often operate below management and that one reason for weak compliance with travel policies is employees' high levels of autonomy over business trips decisions. They prioritise travel time, comfort and convenience over costs and environmental impact. Further, senior staffs and managers might not support the travel policy in practice. As pointed out by Gustafson [26], travel managers often have lower hierarchical positions and lower status than many of the travellers whose travel they are supposed to manage. Lo et al. [17] found that social norms and managerial control were more important in determining business travel frequency and mode choice than commuting travel mode choice.

Although organisations differed on travel policies for business travel and how these are applied in practice, there were examples indicating that a well-established travel policy, incentives that promote cycling and public transport in combination with restrictions on car travel, and the development of an organisational culture that promotes sustainable business travel, creates the favourable conditions for employees to make more sustainable business trips, or to choose a virtual meeting alternative.

*6.3. Methodological Reflections*

Although the offer to participate in the study was given to a relatively large sample (*n* = 525), the proportion of participants who used the MSA and completed the surveys was too small to adequately analyse the quantitative effects of changed travel behaviour as a result of the MSA. The participants who did use the MSA and completed the surveys (*n* = 35) were already travelling more with public transport compared to the non-users. This was also true for the respondents in the interviews. One weakness was that the researchers in the project were not the ones who introduced the MSA to the participants, but it was done by practitioners linked to the organisations. This reduced the ability to control the representability of respondents. The share of MSA users (18% of the respondents making business trips) is in line with the theory of diffusion of innovation estimating the segments of innovators and early adopters to 20% [59], indicating a bias in the analysed sample towards these groups. The problems with attrition and self-selection bias in app-based intervention research have been raised in another paper of this issue [60]. As put forward by these authors, it is challenging to retain participant's interest over time, especially for app-based interventions that suffer from higher dropout rates than other interventions due to a gradual loss of interest in new applications. Moreover, self-recruitment of participants "tends to raise interest in already motivated subgroups of the general population, frequently individuals with high environmental awareness and pro-environmental attitude who may even have already adopted sustainable consumption patterns . . . " [60]. The participants in this study

were recruited from organisations, which is usually an advantage considering that management can encourage participation and thus increase both the number of participants and the heterogeneity of these. In the current case, the engagement of management differed between organisations and a lesson is thus to secure their involvement to increase participation rates. The issue of dropouts was evident in this study as well; only 35 of 193 respondents used the MSA for the full test-period. The dropout was also due to the fact that the MSA had technical weaknesses, and that some participants already had sufficient tools to manage their business travel. Therefore, future studies should carefully consider what type of information and incentives could increase participation.

## 7. Conclusions

The evaluation of a new MSA to support sustainable business travel indicates that the intervention was mostly used for local and regional public transport trips and that it may have made it easier to travel by public transport. When implementing a new system, it is essential to take into account factors that can be influenced by both the intended user and the organisational context in which the system is intended to be implemented. Specifically, we present three factors that affect the success of a new MSA as a means of increasing sustainable business trips: management control and proactiveness; perceived improvement of intervention, and; functions and technical sufficiency. The results highlight the crucial role of management that should take responsibility for establishing a sustainable travel policy, making sustainable transport accessible and creating an organisational culture that encourages environmentally friendly behaviour.

**Author Contributions:** Conceptualization, L.W.H., J.B., S.F. and P.A.; Formal analysis, A.A., L.W.H., J.B., S.F. and P.A.; Methodology, A.A., L.W.H., J.B., S.F. and P.A.; Writing—original draft, A.A. and L.W.H.; Writing—review & editing, A.A. All authors have contributed substantially to the work reported. All authors have read and agreed to the published version of the manuscript.

**Funding:** This research was funded by Vinnova (The Swedish Governmental Agency for Innovation Systems), grant number 2016-03357.

**Acknowledgments:** We wish to acknowledge the work of all project partners as well as the project participants for their contributions to the project. Especially, we thank all the researchers involved in the project who do not figure among the authors of this paper, as well as the supporting institution, Samtrafiken, that helped us throughout the project.

**Conflicts of Interest:** The authors declare no conflict of interest.

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
