# Peer review of "Evaluating a Mobility Service Application for Business Travel: Lessons Learnt from a Demonstration Project"

_sustainability, doi:10.3390/su12030783_

Round 1

Reviewer 1 Report

After review process of article titled “Evaluating a Mobility Service Application for business travel: lessons learnt from a demonstration project” following conclusions were drawn:

The article presents the results of a survey regarding the effectiveness of using a mobility service application to facilitate booking and handling of business trips (in particular public transport) to promote sustainable business travels.  By the theme addressed the article corresponds to the areas of interest of the Sustainability journal.

Even if the structure and content of the article is good from a "technical" point of view, the lack of survey details reduces much of its “scientific” quality.

It is necessary to complete the article with more details regarding the demographic character of the respondents/survey, the correlation of the answers obtained between different characteristics of the interviewees, the insertion of histograms or the frequencies of answers (for e.g.), tables a.s.o. This is necessary because the data obtained (the results) before and after using the MSA tool (presented in tables 4-6) are quite close in value and do not necessarily show the benefits and / or sustainability of using such an instrument.

As a not mandatory request, but maybe it is easier for the reader to read the (%) sign instead of the word "percent" (also, please see and correct the word at page 10 line 361)

For these above presented reasons, the reviewer considers that the necessary completion/additions to the article must be made and the recommendation is "major revision"

Author Response

Thank you for your constructive feedback and suggestions for improvements.

We have supplemented the demographic data with information on respondents' access to a bicycle pool and car pool at the workplace, as well as whether they have a company car. Unfortunately we do not have more information about the respondents as we had to cut demographic questions so that the questionnaire would not take too long to answer. We agree that there is reason to add more information about the quality of the data in relation to the assumptions that follow the singificance tests performed between the variables found in both the pre- and post-study. In fact, we found normality problems with one of the variables, which we therefore excluded from the analysis. There is now a more detailed description of this in the method section. We want to point out that because of the small sample size, we try not to put so much weight on the quantitative part. Instead, we draw most conclusions from the qualitative part. This has now been clarified in several places in the draft, especially in the last sentence of section 5.1.5. We have changed to % characters. We have highlighted our changes in red in the new draft.

Reviewer 2 Report

Dear authors,

This is a pretty good and well-written paper. I only have minor comments:

I was wondering to what extent MSA differs from MaaS? MaaS has received a lot of attention over the past years, but I'm not familiar with MSA. Does MSA only include bussiness travel?

Section 1, lines 66-77. Be aware that some studies have indicated that MaaS might not be as great as often assumed. Some studies show that MaaS has substantial limitations. See, for instance:

Storme, T., De Vos, J., De Paepe, L., Witlox, F., 2019. Limitations to the car-substitution effect of MaaS. Findings from a Belgian pilot study. Transportation Research Part A. In press. https://doi.org/10.1016/j.tra.2019.09.032  

Pangbourne, K., Mladenović, M.N., Stead, D., Milakis, D., 2019. Questioning mobility as a service: Unanticipated implications for society and governance. Transportation Research Part A.  In press. https://doi.org/10.1016/j.tra.2019.09.033

Section 5.1.3: line 361: "percenT"

Section 5.1.4: line 367: Full stop is missing

Author Response

Thank you for your comments!

Because the mobility service evaluated was not a commercial product with a subscription service, and that it was not in the form of a mobile application without a website, we did not think it qualified for the definition of MaaS. Instead, we chose to call it a mobility service application, which we have seen in some other contexts to describe a service that facilitates travel for a specific group, in our case business travelers. Thank you for that suggestion and for the references. We've added a sentence about the possible limitations of MaaS in the introduction of the manuscript. The small errors have been fixed. We have marked the changes made in the text in red.

Reviewer 3 Report

Researching the effectiveness of applications such as MSA is a need for time. As the authors themselves point out, the research of the application has not significantly changed the behaviour of users. It can be influenced by many factors. Certainly the small number of people surveyed makes it impossible to confidently identify groups of users for whom MSA is most useful. However, the presented analysis is an important research step in the monitoring of similar applications, it gives a lot of recommendations for the preparation and organisation of extended research

Author Response

Thank you for your encouraging comments. We really hope for more research going forward with larger populations enabling more robust quantitative analyzes.

Reviewer 4 Report

This is an interesting article, which explains the variable attitude of users towards a new Mobility Service Application. I find very interesting the conclusion that top management must adopt virtuous behaviour in companies if the staff do the same.

Some remarks. Line 22: and;      and

Table 1 social influence: the meaning is not clear.

Table 4: please better illustrate the meaning of the statistical test. In the table the sum of the percentages of the MSA users is 101, the sum of non-users is 99. Please correct for a better understanding.

Table 5 and 6: please better illustrate the meaning of the statistical test.

Author Response

Thanks for your comments. We've fixed the errors you pointed out, highlighted in red in the new draft.

Round 2

Reviewer 1 Report

No further comments